# Simulating a Hockey Hub COVID-19 Mass Vaccination Facility

**DOI:** 10.3390/healthcare10050843

**Published:** 2022-05-04

**Authors:** Ali Asgary, Hudson Blue, Felippe Cronemberger, Matthew Ni

**Affiliations:** 1Disaster and Emergency Management Area and Advanced Disaster, Emergency and Rapid-Response Simulation (ADERSIM), School of Administrative Studies, York University, Toronto, ON M3J 1P3, Canada; hblue@rogers.com (H.B.); fcronemberger@alumni.albany.edu (F.C.); 2Technology Modernization Branch, Innovative Client Service Department, Ottawa, ON K1P 1J1, Canada; matt.ni@ottawa.ca

**Keywords:** mass vaccination, COVID-19, agent-based modeling, hockey hub model, simulation

## Abstract

Mass vaccination is proving to be the most effective method of disease control, and several methods have been developed for the operation of mass vaccination clinics to administer vaccines safely and quickly. One such method is known as the hockey hub model, a relatively new method that involves isolating vaccine recipients in individual cubicles for the entire duration of the vaccination process. Healthcare staff move between the cubicles and administer vaccines. This allows for faster vaccine delivery and less recipient contact. In this paper we present a simulation tool which has been created to model the operation of a hockey hub clinic. This tool was developed using AnyLogic and simulates the process of individuals moving through a hockey hub vaccination clinic. To demonstrate this model, we simulate six scenarios comprising three different arrival rates with and without physical distancing. Findings demonstrate that the hockey hub method of vaccination clinic can function at a large capacity with minimal impact on wait times.

## 1. Introduction

The state of the COVID-19 pandemic changed with the development of several vaccines which have been shown to be effective in reducing the transmissibility and severity of the COVID-19 virus and possibly bringing the pandemic to an end [1]. As more vaccines have become available and more people become eligible for vaccination, a large number of people have needed vaccination in a short period of time [2]. Given the scale of this operation mass vaccination clinics need to be created and operated at high efficiency to achieve the maximum throughput. There are several potential ways that a mass vaccination clinic can be designed and operated. Past research has shown that computer modeling techniques can be used to simulate mass vaccination clinics [3]. These simulations can provide critical analytical data to assist decision makers when considering the best method of operation for mass vaccination clinics. 

The main objective of this paper is to present a model which has been developed to simulate one such method of mass vaccination clinic operation. This method is known as the hockey hub clinic method and presents several advantages over other clinic designs. Key advantages reported include reduced inter-vaccine recipient contact, increased staff efficiency, better recipient safety and reduced wait times [4]. The model itself was built using the AnyLogic software and uses agent-based and discrete event simulation methods to simulate the process of individuals moving through the vaccination clinic. To ensure that the movement of the individual agents through the facility reflects real vaccination clinics, several parameters are considered to determine the agent behavior. These include arrival rate, adverse reaction rate, and average time required in each phase of the vaccination process. To demonstrate this model, we simulated six scenarios which represent a range of intake volumes to examine how varied arrival rates will impact this facility.

## 2. Background and Literature Review

Over the course of the COVID-19 pandemic several infection control measures have been implemented in an attempt to reduce the spread of the disease [5,6]. While these measures have helped to limit disease spread to some extent, vaccination is the most effective and long term solution for community safety and reduction of spread [7,8]. For the benefits of vaccination in society to be fully realized a population must reach the level of herd immunity [7,9]. Herd immunity refers to the point at which there are too few individuals within a population who are susceptible to a disease for it to be able to spread. In order to reach this point, a minimum of approximately 67% of the population must be vaccinated [10]. In Canada, with a population of roughly 38 million, that means that roughly 25 million people would have to be vaccinated [11]. While the country is on the way to reaching this target [12], vaccination efforts need to continue.

Under normal circumstances, vaccination is done through pharmacies and medical clinics [13,14]. These facilities are designed to operate under normal public health conditions, and thus do not have the capacity and facilities necessary to safely conduct mass vaccinations during disease outbreak events [3,15]. That creates a need for specifically designed mass vaccination clinics to be quickly set up and made operational, an effort that requires proper design of a deployment strategy [16,17] as well as local preparedness, which involves understanding demographic, social and contextual factors [18]. 

There are several methods for establishing these kinds of clinics. Drive-through clinics and walk-through clinics have both been adopted as potential solutions to this problem [3]. In both of these clinic types, the individual getting vaccinated moves through a series of static stations where they complete part of the vaccination process. These steps typically include queuing, registration, vaccination, observation, and discharge or exiting [3]. In a drive-through or walk-through model, each of these stages is separate with the individual moving between them either in their car or on foot. For a vaccination clinic to operate efficiently, the time spent in each of these stages should be kept as low as possible. This is not possible in some stages such as recovery and observation because this requires a predetermined amount of time. However, the time spent in stages such as queuing, and registration should be reduced as much as possible to optimize throughputs of vaccination initiatives [19]. 

An alternative method for setting up mass vaccination clinics involves examining high-capacity facilities [20] such as the hockey hub. In the hockey hub model, clients do not walk or drive between the different stages of the vaccination process. Instead, after registration, clients are sent to an individual cubicle where they remain throughout the vaccination and observation process. Healthcare workers administer vaccines by moving between the different cubicles. Vaccine recipients then remain in place and are observed for a set period of time. After observation, each client is discharged from the vaccination facility with the appropriate paperwork. This way clients only need to move twice during their time in the clinic. By moving between clients’ healthcare cubicle, staff can more efficiently distribute vaccines because time is not lost between clients [4]. This method is also beneficial because clients do not need to move between vaccination and observation. As a result, there is less strain put on clients immediately after they are vaccinated, which may help reduce immediate adverse reactions. Another benefit of the hockey hub method is that these clinics can be easily set up on recreation facility hockey rinks that are not in use. That means that the facilities to house this type of clinic are widely available across Canada and could be potentially repurposed.

The effectiveness of this method is demonstrated through its adoption in several public health units around Ontario, including Ottawa, Grey Bruce, and York Region. The model discussed in this paper was used to inform the implementation of the hockey hub clinics in both Grey Bruce and Ottawa and sheds light on the parameters and conditions that should be taken into consideration when undertaking similar endeavors.

## 3. Materials and Methods

To simulate a hockey hub vaccination clinic, an agent-based and discrete event simulation tool was created. AnyLogic (version 8.7.3) modeling and simulation (M&S) software was used as a platform. This simulation tool considers the time that an individual must take at each stage in the vaccination process. Specifically, these stages are queuing, screening/intake, vaccination, observation, and discharge, including the time required to travel between these stages. The simulation tool also accounts for a certain proportion of the population that suffers adverse effects from vaccination. This is done by randomly selecting agents at a set rate and sending them to an adverse reaction stage where they will remain for a set amount of time. Once the time has elapsed, the agents rejoin the rest of the agent population and are discharged from the vaccination clinic. The state chart which defines the way that the agents move through the facility can be seen in Figure 1. In this simulation there are two agent types: clients and vaccinators. Clients are considered vaccinated once they have had the appropriate length of contact with a vaccinator agent.

Figure 2 shows the physical layout of a hockey hub vaccination facility. In the figure, the cubicles where clients receive their vaccines can be clearly seen in the area labeled “immunization”. Queuing and admission are also visible in the area labeled “check-in”. Discharge and the area for recovery of clients with adverse effects can be seen in the remaining area. The simulation tool can also display the layout and agent movement in 3D (see Figure 3).

To determine the conditions of the simulation, several parameters are used in the simulation. Those are listed in Table 1 along with the parameter values used in the base model. The base parameter values were provided by the City of Ottawa based on their experimental run of a hockey hub clinic. However, the simulation tool allows the users to change the parameter values and examine the results accordingly.

As previously mentioned, this simulation tool was developed to examine the implementation of a hockey hub model in the City of Ottawa as a potential method of operating their mass vaccination clinics. The simulation provided information for the adaptation and operation of such a clinic for mass vaccination. The scenarios simulated in this paper are based on situations that would likely be encountered in a large regional center such as Ottawa. 

In this paper six scenarios are presented. These scenarios are meant to demonstrate how the hockey hub clinic will perform under different arrival volumes. 

## 4. Results

Six scenarios were run in order to examine how arrival rates impact this facility. This is also a way of demonstrating the type of information the model can generate and the arrival rates at which the facility runs most efficiently. That means that clients should be arriving at a rate where medical staff are able to constantly vaccinate clients and client wait times are kept to a minimum. The simulation scenarios represent clients arriving at 50, 100 and 150 per hour with and without physical distancing measures for 12.75 h or 765 min. The parameters used in these scenarios are shown in Figure 3 with the physical distancing and arrival rate parameters being altered to create the scenario. For each of these scenarios four metrics were recorded. They are the (1) total number of people vaccinated; (2) average wait times; (3) average time in the clinic; and (4) the total number of people in the clinic at any given time.

Beginning with the total number of people vaccinated in the simulated clinic, the first scenario with 50 arrivals per hour produced a total of 636 and 601 vaccinations with and without physical distancing. This increased to 1314 and 1331 vaccinations with 100 people arriving per hour and to 1926 and 1920 with 150 people arriving per hour (see Figure 4). This result suggests that physical distancing measures do not have a large effect on the number of people who can be vaccinated in this type of facility per day.

In terms of average wait times, it was found that in the scenarios both with and without physical distancing measures, the average wait times were nearly identical. Across all six scenarios, the average wait time did not vary by more than one minute (see Figure 5 and Figure 6). This suggests that even with 150 clients arriving per hour (one every 24 s) the facility has not begun to experience any backlog in the number of clients that can be accepted. It also suggests once again that physical distancing does not on its own increase wait times experienced by clients.

As far as wait times are concerned, minimum and average times spent in the clinic were nearly the same in all scenarios (see Figure 7). All average times were 34 min except one (150 arrivals per hour with physical distancing), which had a mean of 35 min. This trend was repeated for the maximum time spent in the clinic with all non-physical distancing scenarios, and the first two physical distancing scenarios all displaying maximum times between 49 and 51 min. The outlier of this trend is the scenario with 150 arrivals per hour and physical distancing measures in place, which showed a maximum time of 70 min. It is not entirely clear why this scenario had a noticeably longer wait time. One possible answer is that the delay was due to a situation where several clients went to exit the facility at once, a process which was slowed by physical distance measures in place.

Finally, Figure 8 and Figure 9 show the number of people in the clinic during each minute of the simulation. For both the physical distancing and non-physical distancing scenarios, the total number of people in the clinic at any given time generally increased with the increased arrival rate. One interesting note on these findings is that in all scenarios the number of people stayed relatively consistent across the scenario rather than gradually increasing. This suggests that the clinic functions in its maximum capacity even with 150 people arriving per hour. It is also relevant to note that physical distancing does not seem to have reduced the number of people in the clinic at any given time, despite clients needing to be more spread out. 

## 5. Discussion

The main goal of this paper is to present a modeling tool which simulates the behavior of individuals moving through a hockey hub clinic. The findings presented demonstrate that this tool can be used to understand how different arrival rates may affect clinical operations. This can be used by public health mass vaccination administrators and other decision makers because it can allow them to test capacity and clinic operation scenarios quickly and without putting people at risk. This also allows for more informed decisions regarding staff allocation, and public appointment scheduling. Furthermore, by adjusting the parameters of the simulation, public health officials would be able to account for the characteristics of different vaccines such as differing adverse reaction rates or administration times. 

Findings also demonstrate that the hockey hub method of vaccination clinic operation can function at a large capacity with minimal impact on wait times. Even with 150 arrivals per hour (one every 24 s), the average wait time of the clinic remained constant. This robustness was also demonstrated in the findings that physical distancing does not cause any notable decrease in the efficiency of the clinic operation, a finding that may suggest that hockey hubs are well spaced to accommodate a high transit of people with lower risk of an occasional exposure. While it was specifically found that introducing physical distancing measures did not increase the average time that clients have to wait to receive their vaccine, they did increase the maximum wait time experienced by clients. Based on observations of the two and three dimensional representations of the clinic in the model, such an increase is likely due to the traffic flow challenges that physical distancing introduces. Thus, it is important that the impact of physical distancing measures on the flow of clients through the clinic is considered by public health officials who are planning vaccination clinic operations. Another important finding is that, as the arrival rate increased, the number of people in the clinic at any given time also increased. While this result is intuitive, and despite the relatively minor effect of physical distancing to the flow observed, it demonstrates that arrival rates must be managed in order to mitigate the risk of disease spread within the clinic as more people in the clinic means that disease spread is more likely. Despite this challenge, with 150 hourly arrivals per clinic, a city of one million people operating six of these clinics could vaccinate their entire population in 2.8 months provided there were no other obstacles.

Beyond those specific findings, this model illustrates how agent-based and discrete event simulations can be used to help make quick and accurate decisions on disaster response. The model presented here has been used by both the Ottawa and Grey Bruce public health teams to inform their mass vaccination strategy, a fact that demonstrates real-world applications of rapid simulation development can be actionable. Insights from the model include defining the ideal capacity the clinics should have when scheduling vaccinations and related adjustments in terms of resources and preparedness. 

## 6. Conclusions

By using real-world information in the development of a theoretical clinic, models such as this one are able to influence decisions regarding public health risks and issues in a pandemic response. For the special case explored in this paper, the model also demonstrates some key characteristics of the hockey hub vaccination clinics, which may lead to more effective vaccine administration endeavors. Among advantages, the relatively high capacity of this type of facility, and the ability to operate with minimal wait times even while extremely busy, deserve attention. While modeling can be a useful tool in emergency management, further development and refinement is always needed. Future versions of the model could potentially introduce new parameters such as parking space and conditions as well as other customizations involving varying clinic sizes. Those possibilities could make the modeling tool more flexible, and as a result, more broadly useful.

## Figures and Tables

**Figure 1 healthcare-10-00843-f001:**
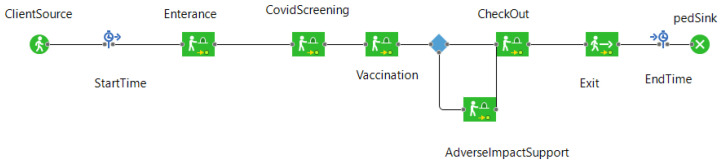
State chart that determines the movement of the agents through the different stages of the vaccination clinic.

**Figure 2 healthcare-10-00843-f002:**
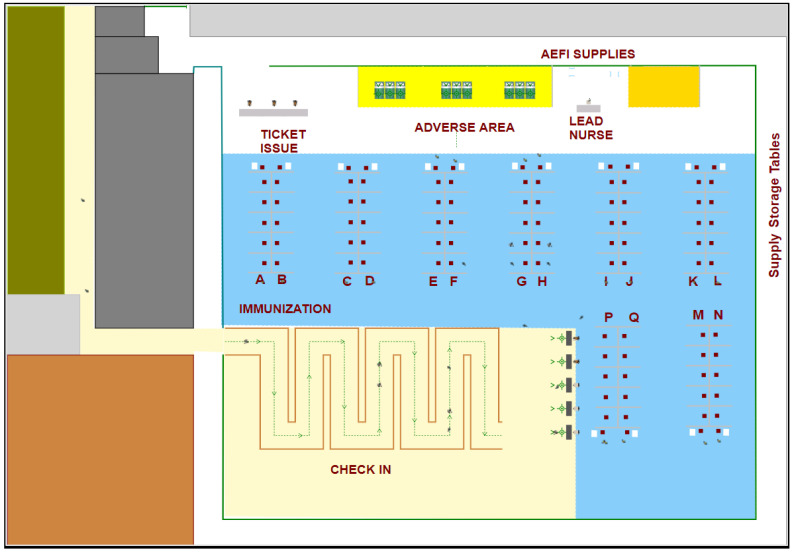
2D physical layout of a hockey hub vaccination clinic in two dimensions.

**Figure 3 healthcare-10-00843-f003:**
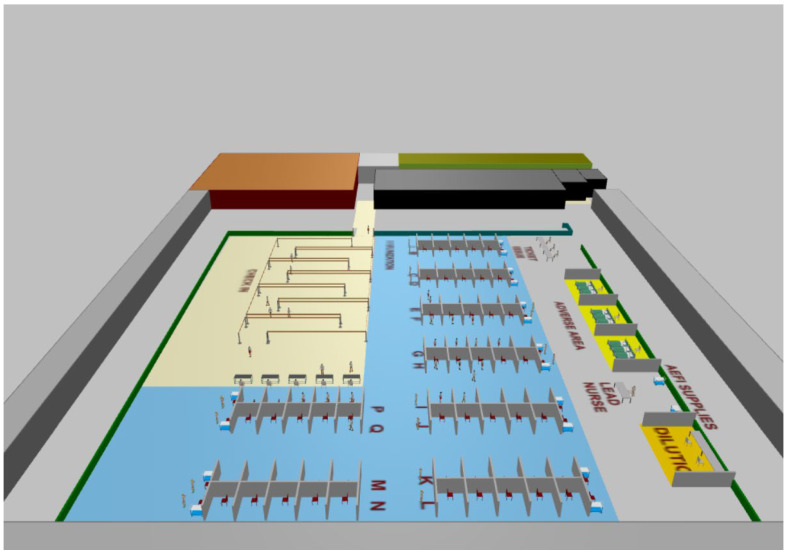
3D physical layout of a hockey hub vaccination clinic in three dimensions.

**Figure 4 healthcare-10-00843-f004:**
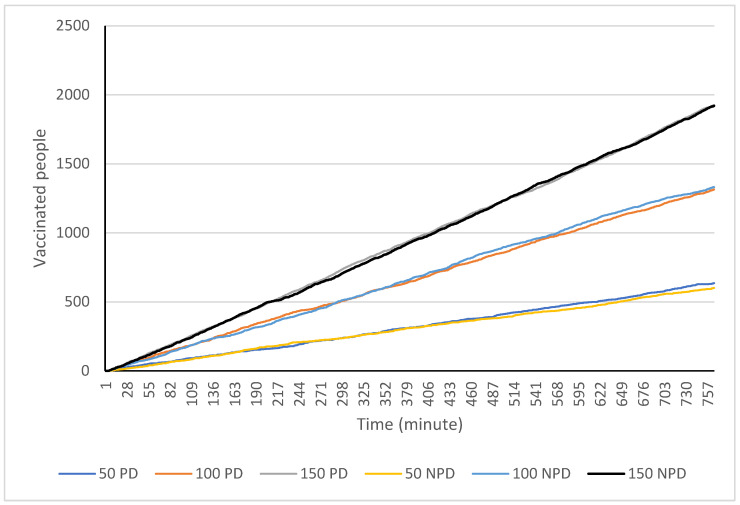
Total number of people vaccinated over the course of 12.75 h at the simulated clinic with and without physical distancing measures in place.

**Figure 5 healthcare-10-00843-f005:**
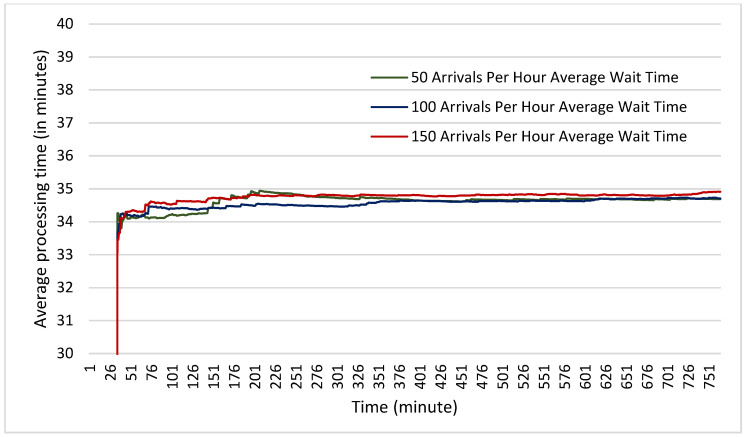
Average time spent waiting by clients at each of the arrival rates for a facility with no physical distancing measures measured in minutes.

**Figure 6 healthcare-10-00843-f006:**
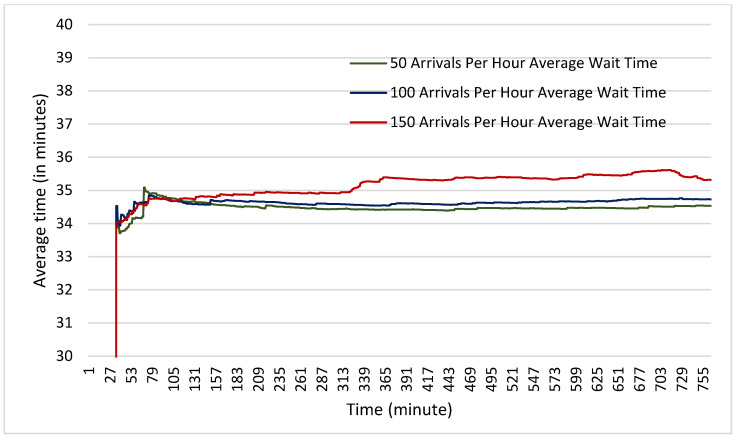
Average time spent waiting by clients at each of the arrival rates for a facility with physical distancing measured in minutes.

**Figure 7 healthcare-10-00843-f007:**
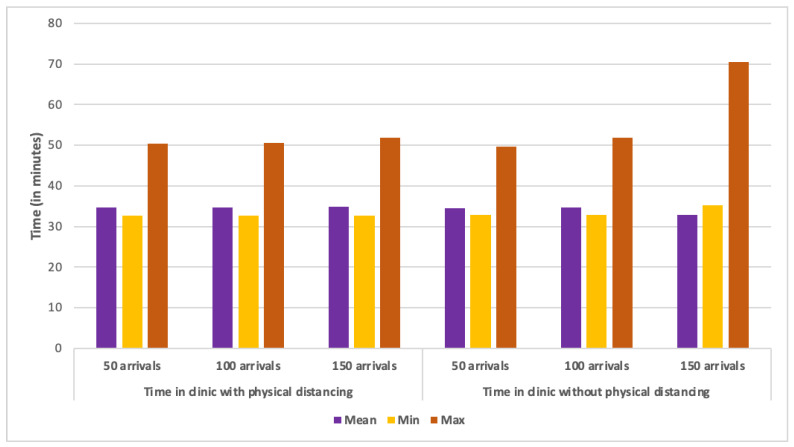
Mean, minimum and maximum time spent by clients in the clinic in all three arrival rate scenarios with and without physical distancing represented in minutes.

**Figure 8 healthcare-10-00843-f008:**
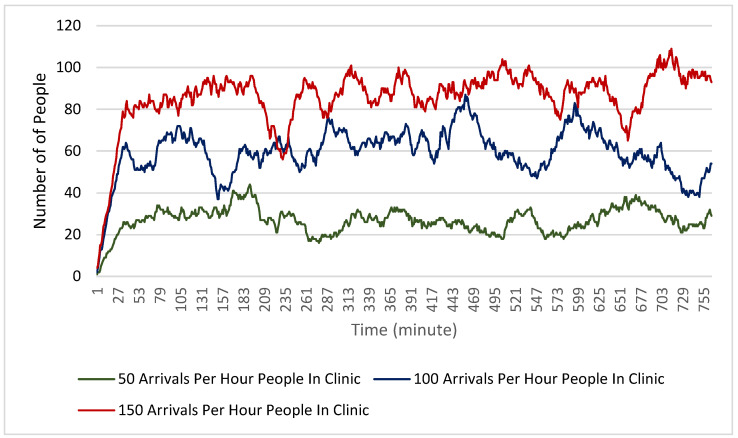
Total number of people in the clinic at each minute during the simulation for each of the arrival rate scenarios with no physical distancing.

**Figure 9 healthcare-10-00843-f009:**
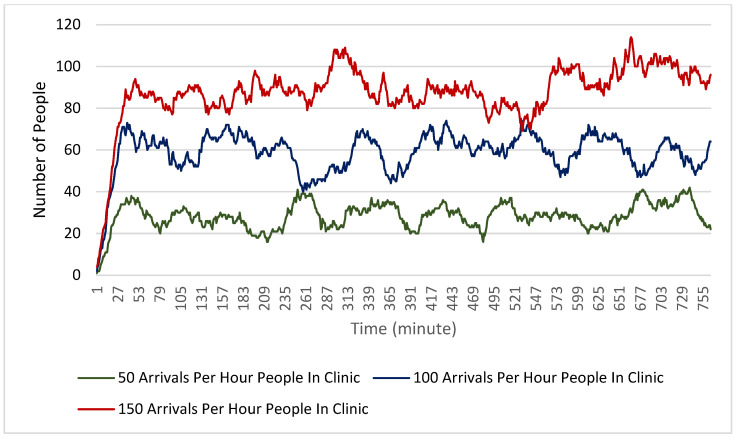
Total number of people in the clinic at each minute during the simulation for each of the arrival rate scenarios with physical distancing.

**Table 1 healthcare-10-00843-t001:** Parameter categories that determine the conditions of the simulation and the corresponding parameter settings of the base model.

Parameter Category	Base Parameter Setting
Adverse Reaction Rate	0.02
Adverse Reaction Time (min)	15
Vaccination Time (min)	27
Screening Time (s)	120
Arrival Rate (per hour)	150
Physical Distancing (True, False)	True
Physical Distancing Distance (m)	2
Ticket (Discharge Timing) (s)	30
Minimum Screening Time (s)	30
Maximum Screening Time (s)	120
Mode Screening Time (s)	90

## Data Availability

Not applicable. AnyLogic simulation model is available upon request.

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
