# Peer review of "Simulating a Hockey Hub COVID-19 Mass Vaccination Facility"

_healthcare, 2022, doi:10.3390/healthcare10050843_

Round 1
Reviewer 1 Report
the manuscript studied the relation between client arrival rate and clinical (hockey hub) operation using model simulation. This study present the impacts of patients arrival rate, plus with and without social distancing, on the effectiveness of mass clinical vaccination. The information provided from this study results can be used by public health mass vaccination administrators and other decision makers to deal with pandemic outbreak. It fit into the scope of journal.
Minor:
- Reference style shall stay consistent throughout. Please check journal requirement.
- In figure 1 some legends were missing
- Figure2 fonts of caption keep consistent
Major:
- More information are needed for choosing the parameters in table -1. in the table, a fixed single value was used for each variables during simulation. What are the rational to choose those variables? Considering it is a dynamic fluctuations for patients numbers in real scenario, shall it make more sense that a range of each variables, instead of single value, reflect the variation dynamics.
2 in comparing with current clinical vaccination strategy, what are the challenge issues or limitations of the current? What are the benefit of mass clinical vaccination?
Author Response
Dear Editors,
I hope you are doing well. Attached you will find a revised version of our manuscript now entitled “Simulating a Hockey Hub COVID-19 Mass Vaccination Facility”. Here I explain how we have addressed each of the reviewers’ comments in the new version. We thank you and the anonymous reviewers for their great comments.
Thank you very much.
Kind regards
Ali Asgary
Corresponding author
Reviewer 1
“the manuscript studied the relation between client arrival rate and clinical (hockey hub) operation using model simulation. This study present the impacts of patients arrival rate, plus with and without social distancing, on the effectiveness of mass clinical vaccination. The information provided from this study results can be used by public health mass vaccination administrators and other decision makers to deal with pandemic outbreak. It fit into the scope of journal.”
Response: Thank you for your comments.
Minor:
- Reference style shall stay consistent throughout. Please check journal requirement.
Response: We addressed this accordingly.
- In figure 1 some legends were missing.
Response: We recreated Figure one to better show different components of the model. Please see the new Fig.1
- Figure2 fonts of caption keep consistent.
Response: We reviewed it for consistency. Please see the new Fig.2
Major:
- More information are needed for choosing the parameters in table -1. in the table, a fixed single value was used for each variables during simulation. What are the rational to choose those variables? Considering it is a dynamic fluctuations for patients numbers in real scenario, shall it make more sense that a range of each variables, instead of single value, reflect the variation dynamics.
Response: Thank you for the comment. The table reflects the parameter values for the base model. However, as can be seen in the results section, we have changed some of these based parameters to examine different scenarios. We provided further clarification under the table to reflect this better. Please see Page 5, changes are colored in red.
2 In comparison with current clinical vaccination strategy, what are the challenge issues or limitations of the current? What are the benefit of mass clinical vaccination?
Response: The mass vaccination clinics were essentially used soon after large volumes of vaccines became available to rapidly vaccinate large number of people. The normal clinics were neither equipped nor had the capacity to immunize large number of people, especially for vaccines that require specific refrigerating equipment. We have discussed these in another paper that has been referenced here. Having said that we added a few more words in the introduction section to further elaborated these.

Reviewer 2 Report
Dear authors,
Thank you for your article, which relevant for logistics during a huge vaccinaiton compaign.
Please, consider some doubts and comments for further improvement of the manuscript:
1) Fig.1: What number 709, etc... stands for? Please, provide an explicative legend for them.
2) Table 1: How the parameter of vaccination time was estimated? 27 minuts is not too much? (In COVID-19 in similar real conditions, I remember max 5 minuts in the box). Or this time includes also 15-20 minuts of observation time after?
3) Are the graphs Fig 4 and Fig.5 the same (looks as they are identical)?
4) You may combine Fig. 8 & 9 (just placing different scenarios next in each coln (with less shadow and the same colour, for example) - easier to compare. The lines with a * within the same cathegory between different scenarious that show significant differences would be a plus.
Small hints:
Line 186: a space is missing: "minute(see Figures 186"
Author Response
Dear Editors,
I hope you are doing well. Attached you will find a revised version of our manuscript now entitled “Simulating a Hockey Hub COVID-19 Mass Vaccination Facility”. Here I explain how we have addressed each of the reviewers’ comments in the new version. We thank you and the anonymous reviewers for their great comments.
Thank you very much.
Kind regards
Ali Asgary
Corresponding author
Reviewer 2
Dear authors,
Thank you for your article, which relevant for logistics during a huge vaccination campaign. Please, consider some doubts and comments for further improvement of the manuscript.
1) Fig.1: What number 709, etc... stands for? Please, provide an explicative legend for them.
Response: We clarified with an explicative legend.
2) Table 1: How the parameter of vaccination time was estimated? 27 minutes is not too much? (In COVID-19 in similar real conditions, I remember max 5 minutes in the box). Or this time includes also 15-20 minutes of observation time after?
Response: The parameter values were provided by the City of Ottawa Public Health based on their experimental Run of the Hockey Hub model. Yes, in the Huckey Hub model all stages are combined. 27 minutes was provided to us by the city of Ottawa. We clarified this in the new version of the paper.
3) Are the graphs Fig 4 and Fig.5 the same (looks as they are identical)?
Response: They look identical but have minor differences. We combined them and modified the text to reflect this. Please see the new Fig. 4
4) You may combine Fig. 8 & 9 (just placing different scenarios next in each column (with less shadow and the same colour, for example) - easier to compare. The lines with a * within the same category between different scenarios that show significant differences would be a plus.
Response: Figure 8 and 9 have been combined into one and we checked the manuscript for this and any extra spaces.
Small hints:
Line 186: a space is missing: "minute(see Figures 186"
Response: We checked the manuscript for this and any extra spaces.

Round 2
Reviewer 1 Report
Authors have provided additional information to address the comments. recommend to accept.